# Interrater reliability of novice examiners using A-mode ultrasound and skinfolds to measure subcutaneous body fat

**Dale R. Wagner**[1]*, **Masaru Teramoto**[2]

**1** Kinesiology & Health Science Dept., Utah State University, Logan, UT, United States of America,
**2** Division of Physical Medicine & Rehabilitation, University of Utah, Salt Lake City, UT, United States of America

* dale.wagner@usu.edu

## Abstract

Examiners with minimal training and skill are often called upon to make body composition assessments using field methods. This study compared the interrater reliability of novice examiners for the skinfold (SKF) and A-mode ultrasound (US) methods of body composition assessment. Undergraduate Kinesiology majors (48 males, 32 females) with minimal training took both SKF and US measurements at three sites (males: chest, abdomen, thigh; females: triceps, suprailiac, thigh). Interrater reliability was significantly better for US compared to SKF at the thigh ($ICC_{US} = 0.975$, $ICC_{SKF} = 0.912$) and abdomen ($ICC_{US} = 0.984$, $ICC_{SKF} = 0.693$) for men and suprailiac ($ICC_{US} = 0.978$, $ICC_{SKF} = 0.883$) for women. Additionally, interrater reliability of the US method was superior to the SKF method for the estimate of male body fat percentage ($ICC_{US} = 0.990$, $ICC_{SKF} = 0.862$). The 95% CI was generally narrower for the US method than the SKF method at each site. The interrater reliability of the US method was superior to or equal to the SKF method for measuring subcutaneous body fat when novice examiners took the measurements.

## Introduction

Body composition is considered a health-related component of physical fitness; therefore, it is common practice to estimate body fat percentage (%BF) during health-fitness screenings and assessments [1, 2]. Numerous methods exist for assessing body composition, and the most common methods have been summarized in texts [3, 4] and review articles [5, 6]. Some of these methods are limited to laboratory settings while others are portable for use in field settings. Although the laboratory methods are thought to be more accurate, they are impractical and too costly for use outside of research or hospital settings. Thus, field methods are typically used by those in the fitness industry, public schools, and athletic organizations to measure and monitor the body composition of their clients, students, or athletes.

One of the most common and widely used field methods of body composition assessment is the skinfold (SKF) caliper [7]. In a global survey of body composition practitioners, the SKF caliper was used more than any other body composition method [8]. This method involves

**Data Availability Statement:** The data sets are available from the figshare repository (https://figshare.com/s/776948a5196029dab5d8).

**Funding:** The authors received no specific funding for this work.

**Competing interests:** The authors have declared that no competing interests exist.

pinching a fold of skin and using the calipers to measure the thickness of the fold. An indirect estimate of subcutaneous fat is obtained, and via prediction equations, is used to estimate % BF. In the hands of a skilled technician, this can be a reasonably accurate method for estimating %BF of individuals, with a biological variability of 3.3% [9]. However, the SKF measurement technique requires considerable practice to become proficient. Jackson and Pollock [10] recommended practicing on 50 to 100 clients to develop skill. Interrater reliability is improved with experience and training [11]. One can only speculate on the experience or training of typical personal trainers, athletic trainers, coaches, dieticians, and clinicians who are currently using the SKF method to estimate the %BF of individuals, but it is likely that many (if not most) have not had sufficient training to perfect the SKF technique.

An alternative to SKFs for measuring subcutaneous fat is ultrasound (US). Whereas SKFs provide an indirect measure of fat thickness with a double layer of skin and the compressed, pinched fold, US offers a direct measure of uncompressed fat thickness [12]. High-resolution B-mode US with automated software to measure fat thickness is the recommended method for measuring the subcutaneous fat of elite athletes [12]. However, high-resolution B-mode US devices found in medical clinics are expensive (> $30,000) and not practical for most personal trainers or other examiners working outside of clinical settings. A relatively inexpensive (< $2,500) and user-friendly A-mode US designed specifically for the measurement of subcutaneous fat is a more practical option for fitness professionals who want to conduct body composition assessments on clients. Subcutaneous fat thicknesses from this A-mode device were comparable to observed thicknesses in dissected cadavers [13] and in-vivo B-mode US measurements [14]. The US method for measuring subcutaneous fat, including the technical principles and differences between A-mode and B-mode, was previously reviewed [15]. The interrater reliability for A-mode US was superior to the SKF method for assessing subcutaneous fat when the technicians were experienced with both methods [16]. However, there is no published record of the interrater reliability of A-mode US in the hands of inexperienced testers. Thus, the purpose of this study was to compare the interrater reliability of the A-mode US to the SKF method when novice examiners were conducting the measurements. Given the difficulty that students have mastering the SKF technique, we hypothesized that the US method might yield better interrater reliability. We believe this study has high practical importance because, in reality, many practitioners likely find themselves making body composition assessments without adequate training with either the SKF or US methods.

## Materials and methods

### Participants

Undergraduate Kinesiology majors from several lab sections of a fitness assessment course enrolled in the study. All of the students had about 1 hour of experience with the SKF method from one lab experience in another course the previous semester. None of the students had any previous experience with the US method. Although all students had to participate in the lab as part of the course, they could opt out of having their data included in the study. A study recruitment script explaining this option and the purpose of the study was read aloud in class. Students choosing to have their lab data included in the study signed a written informed consent. The study was approved by Utah State University's institutional review board (protocol #7960).

### Protocol and materials

One 50-min class period was divided to provide instructional information about the methodology and measurement techniques for both the SKF and US methods. Thus, students received

approximately 20–25 min of classroom instruction for each method. The instructional method included a combination of lecture with PowerPoint slides, video clips, and demonstration.

Following classroom instruction, students measured each other in a laboratory setting. Students worked in groups of three; each student was measured by the other two students using both the SKF and US methods. Thus, each student served as both an examiner twice and as an examinee twice. Prior to conducting the SKF and US measurements, heights and weights were measured with students wearing only shirts and shorts. Height was measured to the nearest 0.1 cm with a wall-mounted stadiometer (Seca 216, Seca Corp., Ontario, CA), and weight was measured to the nearest 0.1 kg with a digital scale (Seca 869, Seca Corp., Ontario, CA).

The same three sites were measured with both SKF calipers (Lange, Beta Technology, Inc., Cambridge, MD) and A-mode US (BodyMetrix BX 2000, IntelaMetrix, Inc., Livermore, CA). These sites included the chest, abdomen, and thigh for males, and the triceps, suprailiac, and thigh for females. The anatomical locations of the sites were described by Jackson and Pollock [10]. The students marked each measurement site on each other using hypoallergenic surgical marking pens. Sites were marked on the right side of the body. The instructor or experienced graduate student verified the correct location of the marking and altered it if necessary.

Once marked, students took SKF and US measurements without any feedback from the instructor or graduate assistants. One set of SKF measurements was taken at each site and then repeated. If the two measurements at a particular site were not within 10% of each other additional measurements were taken until two were within 10% of each other; these two were averaged [3]. The second student examiner did not take any measurements until the first examiner was finished. Site-specific US measurements were made according to the manufacturer's instructions. The software identifies the fat-muscle interface (Fig 1) and automatically prompts the examiner to take multiple measurements at each site until there are at least two that are similar. For all of the measurements, graduate assistants recorded the scores so that the student examiners were not told the others' measurements.

The formulas of Jackson and Pollock [17] and Jackson et al. [18] were used to convert the sum of SKFs into body density for men and women, respectively. Subsequently, body density was converted to body fat percentage with the Siri [19] formula. The SKF thickness will always be greater than the US thickness at an individual measurement site because the US provides a direct measure of thickness while the SKF is a fold, or a double layer. Thus, although the same measurement sites were used for both SKF and US, the BodyMetrix software (Body View Professional) automatically converted the sum of the A-mode US measures into %BF using a proprietary equation unavailable to the public, not the previously mentioned SKF and Siri formulas.

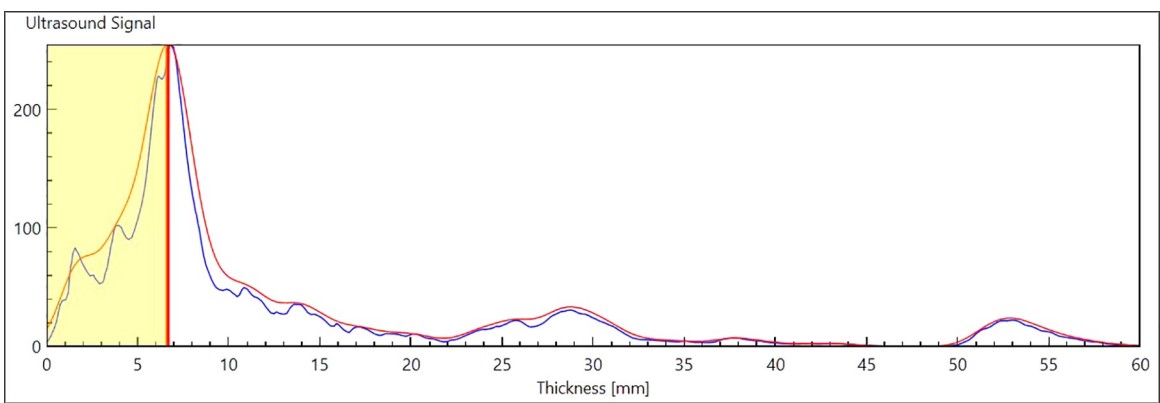

**Fig 1. Example of an A-mode ultrasound graph.** The shaded area represents the subcutaneous fat thickness.

**Table 1. Demographics of participants.**

| Variable | Male ($n$ = 48) | Female ($n$ = 32) |
|---|---|---|
| Age (yr) | 24.4 (1.6) | 22.1 (1.1) |
| Height (cm) | 182.0 (6.9) | 166.7 (5.1) |
| Weight (kg) | 85.5 (14.2) | 68.1 (17.2) |
| Body mass index (kg/m$^2$) | 25.8 (4.2) | 24.4 (5.7) |

Values are mean (SD).

## Statistical analyses

Descriptive statistics were calculated for subject demographics, along with body composition measurements and estimated %BF of the participants by each method (SKF and US). As the measure of interrater reliability, the intraclass correlation coefficient (ICC) [20] and its 95% confidence interval (CI) were calculated for each method at each measurement site and for % BF. Since each subject was measured by different sets of raters (= random effects), one-way random effects to calculate ICCs (Model 1) was used [20]. Additionally, for %BF obtained from two testers for both SKF and US methods, scatterplots were constructed and standard error of measurement (SEM) [21, 22], along with 95% minimal detectable difference (MDD$_{95}$) [23, 24], were calculated in order to examine how %BF scores by two testers were clustered together. Stata/MP 16.0 (StataCorp LLC, College Station, TX) was used for all statistical analyses.

## Results

Complete data were collected on 80 students. Demographics of the subjects are shown in Table 1. Males ($n$ = 48) were, on average, 2.3 years older, 15.3 cm taller, and 17.4 kg heavier than females ($n$ = 32). BMI values were comparable between males and females (25.8 ± 4.2 kg/m$^2$ vs. 24.4 ± 5.7 kg/m$^2$). Females had more subcutaneous fat than did males, as shown by their larger values of SKF and US (Table 2).

Table 3 shows ICCs for each method at each measurement site as well as for %BF. The ICC was significantly higher for the US method compared to the SKF method at the thigh (ICC$_{US}$ = 0.975 with 95% CI = 0.956–0.986, ICC$_{SKF}$ = 0.912 with 95% CI = 0.843–0.950) and abdomen (ICC$_{US}$ = 0.984 with 95% CI = 0.972–0.991, ICC$_{SKF}$ = 0.693 with 95% CI = 0.455–0.828) for males. Additionally, the ICC of %BF for the US method was superior to that for the SKF method

**Table 2. Body composition measurements of participants.**

| | | Method | |
|---|---|---|---|
| | | SKF[a] | US[a] |
| **Males ($n$ = 48)** | Thigh | 15.1 (6.8) | 7.3 (3.3) |
| | Chest | 11.9 (5.8) | 7.5 (4.0) |
| | Abdomen | 18.1 (6.3) | 17.0 (9.6) |
| | %BF | 12.9 (5.0) | 16.4 (6.7) |
| **Females ($n$ = 32)** | Thigh | 28.2 (10.8) | 11.8 (3.5) |
| | Triceps | 22.8 (8.8) | 12.1 (5.3) |
| | Suprailiac | 19.7 (7.5) | 10.8 (5.0) |
| | %BF | 26.4 (7.1) | 26.0 (6.0) |

Values are mean (SD). SKF = skinfold; US = ultrasound; %BF = percent body fat.
[a]Unit in mm except for percent body fat.

**Table 3. Intraclass correlation coefficients for each method at each measurement site.**

|  |  | Method | |
|---|---|---|---|
|  |  | **SKF** | **US** |
| **Males ($n$ = 48)** | Thigh* | 0.912 | 0.975 |
|  |  | (0.843–0.950) | (0.956–0.986) |
|  | Chest | 0.821 | 0.929 |
|  |  | (0.682–0.900) | (0.874–0.960) |
|  | Abdomen* | 0.693 | 0.984 |
|  |  | (0.455–0.828) | (0.972–0.991) |
|  | %BF* | 0.862 | 0.990 |
|  |  | (0.755–0.922) | (0.983–0.995) |
| **Females ($n$ = 32)** | Thigh | 0.922 | 0.832 |
|  |  | (0.842–0.962) | (0.659–0.918) |
|  | Triceps | 0.955 | 0.944 |
|  |  | (0.908–0.978) | (0.886–0.973) |
|  | Suprailiac* | 0.883 | 0.978 |
|  |  | (0.761–0.943) | (0.956–0.989) |
|  | %BF | 0.939 | 0.969 |
|  |  | (0.877–0.970) | (0.938–0.985) |

Values are intraclass correlation coefficient (95% confidence interval). SKF = skinfold; US = ultrasound; %BF = percent body fat.

*Significant difference in intraclass correlation coefficients between skinfold and ultrasound methods.

in males ($ICC_{US}$ = 0.990 with 95% CI = 0.983–0.995, $ICC_{SKF}$ = 0.862 with 95% CI = 0.755–0.922). In females, only the suprailiac site showed a significantly better ICC for the US method than for the SKF method ($ICC_{US}$ = 0.978 with 95% CI = 0.956–0.989, $ICC_{SKF}$ = 0.883 with 95% CI = 0.761–0.943). The differences in ICCs at the other sites and for %BF in females were not significant ($p > 0.05$). Furthermore, the 95% CI was generally narrower for the US method than for the SKF method at each site. According to the scatterplots of %BF obtained from two testers (Figs 2 and 3), the US method generally displayed a better interrater reliability than did the SKF method, as individual data points for the US method were clustered more around the line of identify. SEM of %BF from two testers measured by the US method for males and females were 0.94% and 1.48%, respectively, compared to 2.61% (males) and 2.41% (females) resulting from the SKF method, indicating the super interrater reliability for the US method. Further, $MDD_{95}$ of %BF from the US method for males and females were 2.60% and 4.10%, respectively. The interpretations are: when %BF of a single subject is estimated by two novice raters using the US method, the expectation is that 95% of repeated-measured scores show random variations of less than 2.60% for a male subject and 4.10% for a female subject. These values from the SKF method were 7.25% (males) and 6.67% (females), indicating wider random variations by the SKF method than the US method.

## Discussion

The primary finding from the present study was that the interrater reliability of the US method was equal to or superior to the SKF method in the hands of novice examiners. This occurred despite it being the first experience with the US method for the student examiners. The interrater reliability of the SKF method has been well studied; however, the interrater reliability of the US method was previously unknown for inexperienced testers. The interrater ICC for

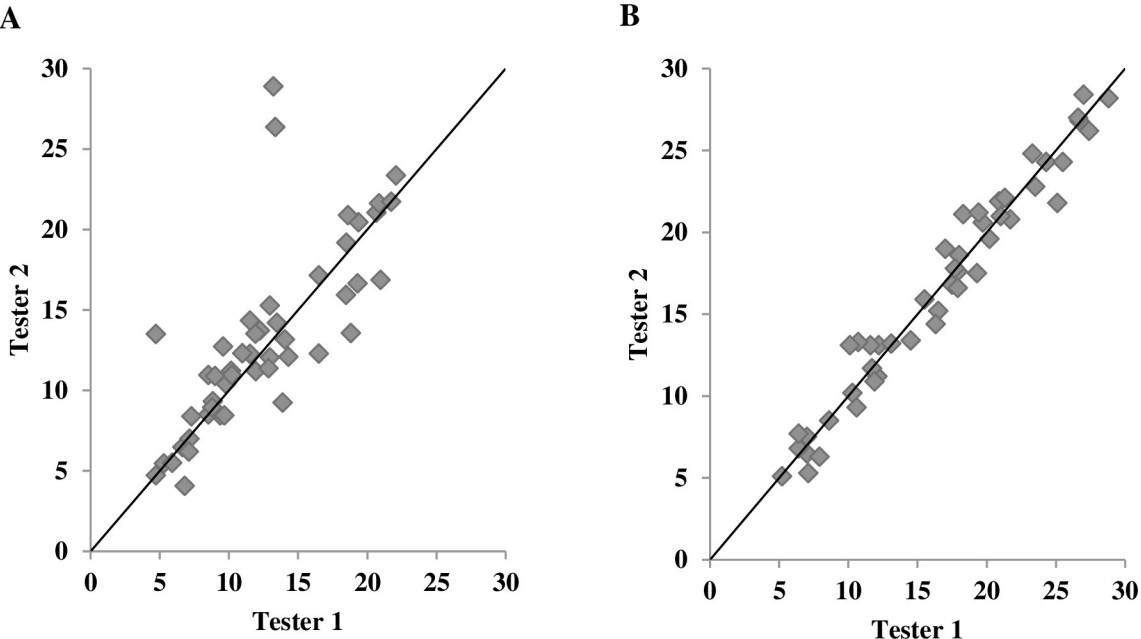

**Fig 2. Scatterplot of percent body fat in males obtained from two testers.** (A) skinfold method and (B) A-mode ultrasound method.

estimating %BF using the SKF method was reported to be 0.97 to 0.99 when experienced examiners were taking the measurements [16, 25]. However, Kispert and Merrifield [26] reported much lower interrater reliability coefficients of 0.62 to 0.85 for individual sites when measured by physical therapy students inexperienced with the SKF technique. Similarly, Kerr and colleagues [11] reported better interrater reliability at each individual measurement site

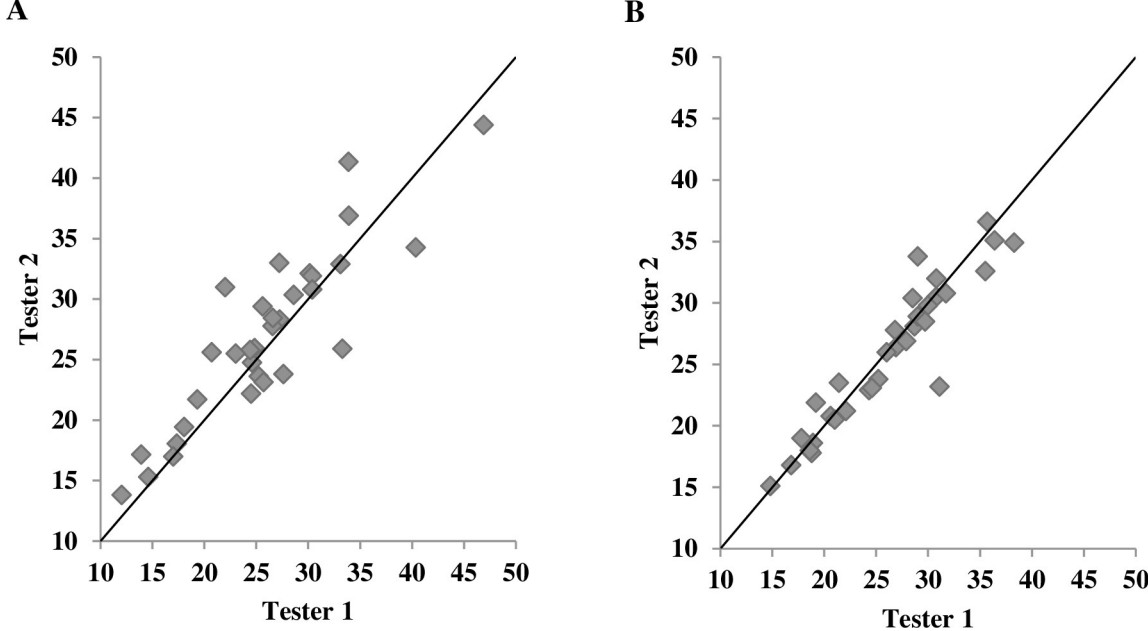

**Fig 3. Scatterplot of percent body fat in females obtained from two testers.** (**A**) skinfold method and (**B**) A-mode ultrasound method.

for technicians who trained 30 min with an expert compared to those who did not participate in the training. The site-specific ICCs for the trained technicians ranged from 0.86 to 0.99 and for the untrained testers from 0.31 to 0.95.

In contrast to the amount of interrater reliability research that has been done with the SKF method, interrater reliability specific to the BodyMetrix BX2000 A-mode ultrasound is almost nonexistent. To our knowledge, only two previous studies examined this. Wagner et al. [16] reported similar ICCs for US (0.987) and SKF (0.966) for the estimate of %BF of collegiate athletes when experienced technicians took the measurements. Despite the similar ICCs, it was noted that the 95% CI was much narrower for the US method (0.976 to 0.993) compared to the SKF method (0.328 to 0.991). Thus, even with examiners experienced in both the SKF and US methods, there was better agreement with the US. Hendrickson et al. [27] recently reported a slightly lower interrater reliability ICC (0.87) for the BX2000 than Wagner et al. [16]; however, they did not report whether or not the technicians were experienced, and there was no comparison to skinfolds. Additionally, their ICC increased to 0.96 in a subset of trauma subjects, and they concluded that the interrater reliability of the BX2000 was excellent.

In general, the ICCs for both the SKF and US measurements taken by the novice examiners in the present study were good (Table 3); they were greater than those reported by Kispert and Merrifield [26] for inexperienced testers, but not quite as large as those reported for experienced examiners [16]. A likely reason for the good ICCs among novice examiners in the present study is that the measurement site locations were marked by the students and then checked by experienced examiners prior to the students conducting the measurements. Previous research suggests that variation in site location is a major source of variability among examiners. For example, Ruiz et al. [28] reported average SKF differences of about 2.5 mm when the caliper placement varied by 1 inch (2.54 cm). Hume and Marfell-Jones [29] created a 1-cm grid pattern around specified measurement sites. They reported that measuring only 1 cm away from the defined site location produced significant differences in the majority of SKF measurements. Thus, with the variability of site location removed, the present study can be regarded as an evaluation of the interrater reliability of the methods and techniques rather than the students' ability to locate the correct measurement sites.

It is important to emphasize that the students had no prior experience taking measurements with the A-mode ultrasound device, and were given only about 25 minutes of classroom instruction regarding the theory and technique of using ultrasound to obtain subcutaneous fat thickness measurements. The software package for the BodyMetrix device automatically selects the peak corresponding to the fat-muscle interface. However, this auto-selection can be manually overridden if necessary. There were likely some instances in which the student examiners probably recorded the software-recommended peak as the fat-muscle interface when they should have overridden the default. For example, the US measurement should never be greater than the SKF measurement because the SKF is a double layer. In a few instances, student examiners recorded US thicknesses that exceeded the SKF measurement, resulting in large errors. Other potential measurement errors include applying too much or inconsistent pressure to the skin when taking the measurement or not measuring in the correct location despite the site being marked. With additional instruction or oversight from an experienced technician during the measurement, these errors could easily be identified and corrected; consequently, even greater interrater reliability of the US method with novice testers is likely with a small amount of additional supervision.

No formal qualitative analysis was performed. However, anecdotally, students commented that the technique for the US measurement was "easier" or preferable to the SKF technique. These qualitative comments coincide with and further support the findings of better interrater reliability for the US method compared to the SKF method.

The findings of this study are limited to the BX2000 A-mode ultrasound and the accompanying software. Other ultrasound-software combinations may not be as easy to use leading to poorer interrater reliability. Buxadé et al. [30] reported inferior test-retest and interrater reliability for A-mode ultrasound compared to skinfold measurements when both methods were applied to eight measurement locations in a heterogeneous sample of 84 adults. However, a different A-mode ultrasound was used (Renco Lean-Meater Series 12), and both technicians were experienced with the ultrasound and skinfold techniques.

It is important to note that the purpose of this study was to evaluate interrater reliability of the methods for novice examiners, not to validate the SKF or US method for estimating %BF. Given individual variation in the ratio of subcutaneous to internal fat, fat patterning, and skin thickness, some experts recommend against estimating total %BF from the fat thicknesses obtained at individual sites [31, 32]. Nevertheless, the SKF method, whether using only individual site measurements or converting SKF data to %BF, continues to be one of the most commonly used field methods of body composition assessment by practitioners [7]. Given the prevalence of the SKF method and the suggestion that the US method could supplant SKFs as a field method for practitioners [16], we believe this evaluation of the interrater reliability of novices using both methods is of practical value.

The practicality and generalizability of these results to real world application were strengths of this study. As mentioned in the introduction, perfecting the SKF technique requires extensive practice [10], and unfortunately, many coaches and fitness trainers may be taking these measurements without the requisite experience. Findings from the present study suggests that this learning curve might be reduced using US rather than SKF, but this hypothesis is yet to be tested. Finally, this study was limited to interrater reliability data only. A follow-up check of the students' measurements by an expert technician would have added a measure of validity.

In conclusion, interrater reliability of novice examiners for measuring subcutaneous fat was better for A-mode US compared to SKF, with significantly greater ICCs and narrower 95% CIs for the US method. With a small amount of additional training or supervision, it is likely that the interrater reliability for the US method would further improve. In both fitness and clinical settings, anthropometric measurements are often taken by different technicians. Unfortunately, some technicians often take these measurements without adequate training. Thus, interrater reliability, particularly among novice examiners, is a source of measurement error. Based on the findings from this study, the US method, rather than the SKF method, can reduce the interrater error associated with measuring subcutaneous fat.

## Acknowledgments

Thanks to teaching assistants, Adrianna Robison and Devin Vance, for their supervision of the labs in which this data collection occurred.

## Author Contributions

**Conceptualization:** Dale R. Wagner.

**Data curation:** Dale R. Wagner.

**Formal analysis:** Masaru Teramoto.

**Investigation:** Dale R. Wagner.

**Methodology:** Dale R. Wagner.

**Project administration:** Dale R. Wagner.

**Resources:** Dale R. Wagner.

**Software:** Masaru Teramoto.

**Supervision:** Dale R. Wagner.

**Writing – original draft:** Dale R. Wagner, Masaru Teramoto.

**Writing – review & editing:** Dale R. Wagner, Masaru Teramoto.

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
