## [Decision Letter · Decision Letter 0]

4 Nov 2020

PONE-D-20-28461

Interrater reliability of novice examiners using A-mode ultrasound and skinfolds to measure subcutaneous body fat

PLOS ONE

Dear Dr. Wagner,

Thank you for submitting your manuscript to PLOS ONE. After careful consideration, we feel that it has merit but does not fully meet PLOS ONE’s publication criteria as it currently stands. Therefore, we invite you to submit a revised version of the manuscript that addresses the points raised during the review process.

Carefully address the points presented by the reviewers to the best of your ability. 

We look forward to receiving your revised manuscript.

Kind regards,

Cherilyn N. McLester, PhD

Academic Editor

PLOS ONE

Journal Requirements:

2. Please include additional information regarding the data collection tool used in the study and ensure that you have provided sufficient details that others could replicate the analyses.

For instance, if you developed a questionnaire as part of this study and it is not under a copyright more restrictive than CC-BY, please include a copy, in both the original language and English, as Supporting Information, or include a citation if it has been published previously.

Reviewers' comments:

Reviewer's Responses to Questions

**Comments to the Author**

1. Is the manuscript technically sound, and do the data support the conclusions?

Reviewer #1: Yes

Reviewer #2: Yes

2. Has the statistical analysis been performed appropriately and rigorously? 

Reviewer #1: Yes

Reviewer #2: Yes

3. Have the authors made all data underlying the findings in their manuscript fully available?

Reviewer #1: Yes

Reviewer #2: Yes

4. Is the manuscript presented in an intelligible fashion and written in standard English?

Reviewer #1: Yes

Reviewer #2: Yes

5. Review Comments to the Author

Reviewer #1: Overall

This is interesting work that has great practical implications for fitness professionals or clinicians. The interrater reliability for two body composition techniques, one being widely used, that are easily accessible for assessing body composition were compared. The manuscript is written well and has great potential for aiding practitioners, but a couple of additional analyses and methodological details would improve the utility of this information.

Abstract

N/A

Introduction

N/A

Methods

Line 81: Considering the nature of this work and importance of familiarity, it would be helpful if there was at least a rough approximation of the average experience (e.g., < 1 hr?) for these students with the SKF technique

What was done to limit compression via the US probe and keep it consistent? It is well known that this influences the morphology of the tissue deep to the probe, so this is at least worth mentioning. Differences in compression could arguably be one of the larges influences on interindividual variability.

A limitation the unknown equation that is used for the US, especially since this means the two techniques are likely using different equations to calculate BF%

For a more comprehensive examination of reliability, the standard error of measurement (SEM) should be provided for both modalities. The SEM provides a different analysis of reliability and is more useful for practitioners. See Weir, 2005.

Given the practical utility of this study and its findings, it would be much improved if the minimal difference to be considered real was calculated (aka MD). This could be very useful for practitioners to know, depending on the testing modality, what a meaningful change would be.

A figure indicating a typical US scan and the selected subcutaneous fat would helpful, especially considered the feasibility for novices to use this technique.

Results

Line 13: “accumulated” indicates there was an intervention or that this was a longitudinal study.

Discussion

Based on a qualitative inspection of the scatterplots, it looks the greatest interindividual differences were at relatively higher fat levels. Was this a common observation? While this is to be expected, it is worth mentioning as it could be useful for practitioners to be aware of.

Reviewer #2: This topic is of important to athletic trainers, personal trainers, coaches etc, as field methods are generally used to measure body composition. Overall, it is a very well designed study and well written.

Introduction:

Does the study have a hypothesis?

Line 45: Need more than one reference to support this statement.

Methods:

Please provide a power analysis. What was the p value set to for significance?

Discussion:

Future research? What are the limitations and strengths of this study?

6. PLOS authors have the option to publish the peer review history of their article (what does this mean?). If published, this will include your full peer review and any attached files.

Reviewer #1: No

Reviewer #2: No

---

## [Author Response · Author response to Decision Letter 0]

16 Nov 2020

A "response to reviewers" file is included.

---

## [Editor Report · Decision Letter 1]

2 Dec 2020

Interrater reliability of novice examiners using A-mode ultrasound and skinfolds to measure subcutaneous body fat

PONE-D-20-28461R1

Dear Dr. Wagner,

We’re pleased to inform you that your manuscript has been judged scientifically suitable for publication and will be formally accepted for publication once it meets all outstanding technical requirements.

Kind regards,

Cherilyn N. McLester, PhD

Academic Editor

PLOS ONE
---

## [Editor Report · Acceptance letter]

4 Dec 2020

PONE-D-20-28461R1 

Interrater reliability of novice examiners using A-mode ultrasound and skinfolds to measure subcutaneous body fat 

Dear Dr. Wagner:

I'm pleased to inform you that your manuscript has been deemed suitable for publication in PLOS ONE. Congratulations! Your manuscript is now with our production department. 

Kind regards, 

on behalf of

Dr. Cherilyn N. McLester 

Academic Editor

PLOS ONE